# MDM2-Mediated p21 Proteasomal Degradation Promotes Fluoride Toxicity in Ameloblasts

**DOI:** 10.3390/cells8050436

**Published:** 2019-05-10

**Authors:** Huidan Deng, Atsushi Ikeda, Hengmin Cui, John D. Bartlett, Maiko Suzuki

**Affiliations:** 1Division of Biosciences, College of Dentistry, The Ohio State University, Columbus, OH 43210, USA; deng.597@osu.edu (H.D.); ikeda.39@osu.edu (A.I.); bartlett.196@osu.edu (J.D.B.); 2College of Veterinary Medicine, Sichuan Agricultural University, Wenjiang, Chengdu 611130, China; cui580420@sicau.edu.cn; 3Department of Oral Biology and Diagnostic Sciences, The Dental College of Georgia, Augusta University, Augusta, GA 30912, USA

**Keywords:** MDM2, p53, p21, ameloblast, fluoride, fluorosis

## Abstract

Fluoride overexposure is an environmental health hazard and can cause enamel and skeletal fluorosis. Previously we demonstrated that fluoride increased acetylated-p53 and its downstream target *p21* in ameloblast-derived LS8 cells. However, p21 function in fluoride toxicity is not well characterized. This study seeks to gain a better understanding of how p53 down-stream mediators, p21 and MDM2, respond to fluoride toxicity. LS8 cells were treated with NaF with/without MG-132 (proteasome inhibitor) or Nutlin-3a (MDM2 antagonist). NaF treatment for 2–6 h increased phospho-p21, which can inhibit apoptosis. However, phospho-p21 and p21 were decreased by NaF at 24 h, even though *p21* mRNA was significantly increased at this time point. MG-132 reversed the fluoride-mediated p21 decrease, indicating that fluoride facilitates p21 proteasomal degradation. MG-132 suppressed fluoride-induced caspase-3 cleavage, suggesting that the proteasome plays a pro-apoptotic role in fluoride toxicity. NaF increased phospho-MDM2 in vitro and in mouse ameloblasts in vivo. Nutlin-3a suppressed NaF-mediated MDM2-p21 binding to reverse p21 degradation which increased phospho-p21. This suppressed apoptosis after 24 h NaF treatment. These results suggest that MDM2-mediated p21 proteasomal degradation with subsequent phospho-p21 attenuation contributes to fluoride-induced apoptosis. Inhibition of MDM2-mediated p21 degradation may be a potential therapeutic target to mitigate fluoride toxicity.

## 1. Introduction

Fluoride is ubiquitous in the environment and is present in bones, teeth and calcified tissues. Fluoride is known as an effective caries prophylactic [1]. The U.S. Public Health Service (PHS) recommends public water fluoridation to prevent caries and the optimal recommended fluoride concentration in drinking water is 0.7 ppm corresponding to 0.04 mM NaF [2]. However, fluoride overexposure can cause acute or chronic health problems. For example, fluoride in volcanic gases and ash can be an environmental health hazard. In 1783, Laki volcanic eruptions in Iceland released highly toxic hydrogen fluoride (HF) that caused severe skeletal fluorosis resulting in the mass death of inhabitants and grazing livestock [3]. Moreover, high fluoride concentration in groundwater can lead to potential fluoride contamination in drinking water [4]. High levels of fluoride have caused health issues worldwide including in the USA, China, India and Africa [5]. These health issues include enamel fluorosis [6], skeletal fluorosis [7], neurotoxicity [8] and reproductive toxicity [9].

Dental fluorosis is a developmental disorder caused by fluoride overexposure during enamel formation. The cells of the enamel organ responsible for enamel formation are ameloblasts. Enamel development (amelogenesis) occurs in stages, pre-secretary, secretary, transition and maturation [10]. Exposure to high fluoride concentrations can cause hypomineralized, mottled, discolored, and porous enamel that is susceptible to decay. The prevalence of dental fluorosis among the population in the USA is increasing and mild to severe dental fluorosis among children (33.4% aged 6–11 and 40.6% aged 12–15) is a concern [11]. Other than avoiding excessive fluoride ingestion, treatment to prevent dental fluorosis remains unknown.

Previously we reported that high dose fluoride causes cell stress, endoplasmic reticulum (ER) stress [12,13] and oxidative stress [14,15] followed by mitochondrial damage, DNA damage and apoptosis [16] resulting in impairment of ameloblast function. The p53 tumor suppressor contributes to the cellular DNA damage response and apoptosis [17]. Following DNA damage, p53 is acetylated at Lys379 (Ac-p53) [18]. Recently, we reported that fluoride induced p53 acetylation [Lys379] in ameloblast-derived LS8 cells in vitro and in rodent ameloblasts in vivo [19]. These results suggest that Ac-p53 plays a critical role in fluoride-induced DNA damage and apoptosis. However, the p53 downstream pathway in fluoride toxicity is not well characterized.

p53 increases transcription of the cyclin-dependent kinase inhibitor 1A/p21 (p21). p21 functions in cell cycle arrest, transcriptional regulation, and anti-apoptosis. These functions are largely dependent on p21 post-translational modification, p21 protein interactions, p21 subcellular localization, and on cell type and specific cellular stresses [20]. Inhibition of apoptosis is the best-known oncogenic function of p21. In the presence of intact p53, p21 can counteract p53-dependent apoptosis. Loss of p21 promotes drug-induced DNA damage and p21 activation protects cells from this damage [21]. Phosphorylation of p21 (p-p21) at Thr145 enhances p21 protein stability and promotes cell survival [22]. This phosphorylation induces p21 relocalization from the nucleus to the cytosol [23]. In the cytosol, p21 can inhibit apoptosis through binding to procaspase 3 to block its proteolytic activation [24].

The ubiquitin E3 ligase murine double minute 2 protein (MDM2) is also a p53 target. p53 increases *Mdm2* expression and MDM2 can inhibit p53 through a negative feedback mechanism [25]. MDM2 binds to p53 and promotes p53 ubiquitin-proteasomal degradation [26]. In contrast, MDM2 also binds to p21, which also increases p21 proteasomal degradation [27]. MDM2 activity is regulated by post-translational modifications, especially phosphorylation. Akt-mediated phosphorylation of MDM2 (p-MDM2) at Ser166 and Ser186 increases MDM2-mediated ubiquitination and degradation of p53 [28]. Recently, it was reported that extracellular signal-regulated kinase (ERK)-mediated MDM2 phosphorylation [Ser 166] promotes p21 degradation [29]. However, MDM2 function in fluoride toxicity remains to be elucidated.

A better understanding of the mechanisms of fluoride toxicity is necessary to identify therapeutic targets that mitigate toxicity. Here, we investigated the crosstalk among p53, MDM2 and p21 in fluoride toxicity and demonstrated that MDM2-p21 binding promotes fluoride-induced apoptosis through MDM2-mediated p21 degradation.

## 2. Materials and Methods

### 2.1. Animals

C57BL/6 mice (6-week-old) were purchased from Charles River Laboratories (Wilmington, MA) and were provided drinking water containing 0 or 150 ppm fluoride for 6 weeks. Then, the animals were euthanized and their incisors were extracted for immunohistochemical analysis [30]. All animals were treated humanely and all handling procedures were approved by the Institutional Animal Care and Use Committee (IACUC) at the Forsyth Institute. The Forsyth Institute is accredited by the Association for Assessment and Accreditation of Laboratory Animal Care International (AAALAC) and follows the Guide for the Care and Use of Laboratory Animals (NRC1996). Note that the fourth and senior authors were employed by The Forsyth Institute through October 2015 when the animal experiments were completed.

### 2.2. Cell Culture

The mouse ameloblast-derived cell line (LS8) was provided by Dr. Malcolm L. Snead [31]. LS8 cells were maintained in alpha minimal essential medium with GlutaMAX (Life Technologies, Grand Island, NY, USA) supplemented with fetal bovine serum (10%) and sodium pyruvate (1 mM). Cells were treated with sodium fluoride (NaF) with/without Nutlin-3a (MDM2 antagonist) or MG-132 (proteasome inhibitor) as indicated. NaF was obtained from Fisher Scientific (Pittsburgh, PA, USA). Nutlin-3a and MG-132 were purchased from Selleck Chemicals (Houston, TX, USA).

### 2.3. Real-Time Quantitative Polymerase Chain Reaction (qPCR) Analysis

Total RNA was extracted from cells using Direct-zol RNA MiniPrep (Zymo Research Corp, Irvine, CA, USA). The cDNA was synthesized using iScript cDNA Synthesis Kit (BioRad, Hercules, CA, USA). The cDNA was subjected to qPCR amplification on a QuantStudio 3 thermal cycler (Thermo Scientific, Rockford, IL, USA). Primer sequences for the mouse are presented in Appendix A. *Gapdh* was used as an internal reference control gene because of its consistent expression with experimental treatments. Data from quantitative polymerase chain reaction (qPCR) were analyzed using the 2^−ΔΔCT^ method [32]. At least three biological replicates were analyzed for each experiment.

### 2.4. Western Blot Analysis

Cells were lysed and proteins were extracted with radioimmunoprecipitation assay (RIPA) lysis buffer (Thermo Scientific) containing protease inhibitor cocktail (Thermo Scientific). Protein concentration was determined by bicinchoninic acid assay (BCA) protein assay kit (Thermo Scientific). Equal amounts of protein sample were loaded into Mini-Protean TGX gels (BioRad) and transferred to nitrocellulose filter membranes. The membranes were blocked in 5% nonfat dry milk or 5% bovine serum albumin (BSA) for 1 h at room temperature (RT), then incubated with the primary antibodies overnight at 4 °C. The primary antibodies were rabbit anti-p53, rabbit anti-acetylated p53 [Lys379], rabbit anti-cleaved caspase 3, rabbit anti-γH2AX, rabbit anti-phospho-MDM2 [Ser166], rabbit anti-ubiquitin, rabbit anti-β actin and mouse anti-β actin (Cell Signaling Technology, Boston, MA, USA), rabbit anti-p21, rabbit anti-MDM2 (Abcam, Cambridge, MA, USA) and rabbit anti-phospho-p21 [Thr145] (Thermo Scientific). The membranes were then washed with Tris-Buffered Saline (TBS)-Tween (TBST) and incubated with the horseradish peroxidase (HRP)-conjugated secondary antibodies; goat anti-mouse IgG (Sigma-Aldrich, St. Louis, MO, USA) or goat anti-rabbit IgG (Biorad) at RT for 1 h. After washing with TBST, enhanced chemiluminescence was performed with SuperSignal West Pico (Thermo Scientific) and the signal was detected by myECL imager (Thermo Scientific). Bands were quantified by MyImage analysis software (version 1.1), Thermo Scientific). At least three biological replicates for each experiment were performed and representative images are shown. Protein expression was normalized by use of the loading control protein (β-actin). Relative protein expression and statistical significance were analyzed by one-way analysis of variance (ANOVA) with Fisher’s least significant difference (*LSD*) post-hoc test using the SPSS statistics 20 software (version 20).

### 2.5. Immunoprecipitation

The co-IP assay was performed using Pierce™ Co-immunoprecipitation Kit (Thermo Scientific), according to the manufacturer’s instructions. The cells were washed twice with ice-cold phosphate-buffered saline (PBS) and lysed with ice-cold IP Lysis/Wash Buffer and incubated on ice for 5 min. The lysate was centrifuged at 13,000× *g* for 10 min at 4 °C. Protein concentration was determined by BCA protein assay kit (Thermo Scientific). 1 mg total protein was used for co-IP with antibodies; mouse anti-p53 (Cell Signaling Technology) and mouse anti-p21 (BD Biosciences, San Jose, CA, USA). Mouse IgG was used as the negative control (Cell Signaling Technology). The immunocomplexes were analyzed by Western blotting with antibodies; rabbit anti-MDM2, rabbit anti-p21 (Abcam), rabbit anti-p53, and rabbit anti-ubiquitin (Cell Signaling Technology). 30 μg protein was used for input. At least three biological replicates for each experiment were performed and representative images are shown.

### 2.6. Immunocytochemistry

Immunocytochemistry was performed to detect p-p21 in vitro. LS8 cells were cultured on micro cover glasses (VWR, Radnor, PA, USA) in 24-well plates and treated with NaF for 6 h or 24 h. After that, cells were washed with PBS and fixed with 4% PFA at RT for 10 min. Then, cells were incubated with 0.1% Triton X-100 at RT for 10 min followed by blocking with 1% BSA at RT for 1 h. Next, cells were incubated with rabbit anti-p-p21 [Thr145] antibody (Thermo Scientific) and mouse anti-β actin antibody (Cell Signaling Technology) overnight at 4 °C. After washing with PBS, cells were incubated with secondary antibodies; AlexaFluor 488-conjugated goat anti-rabbit IgG (Cell Signaling Technology) and AlexaFluor 594-conjugated goat anti-mouse IgG (Thermo Scientific) at RT for 1 h. Then, cells were incubated with DAPI (Thermo Scientific) at RT for 5 min. Cells were analyzed using fluorescence microscopy (PhotoFluor LM-75, 89 North, Burlington, VT, USA). At least three biological replicates for each experiment were performed and representative images are shown.

### 2.7. Immunohistochemistry

Mouse incisors were extracted after fluoride treatment for 6 weeks and fixed in paraformaldehyde, demineralized with EDTA for 2 weeks, and embedded in paraffin. Sections were incubated with primary antibodies: rabbit anti-p-MDM2 [Ser185] (Thermo Scientific) followed by incubation with a peroxidase-conjugated secondary antibody, Vectastain ABC Regent (Vector Labs, Burlingame, CA, USA), and DAB kit (Vector Labs). Then, sections were counterstained with 0.1% Fast Green in PBS and examined by light microscopy.

### 2.8. Statistical Analysis

The qPCR results and Western blot results were analyzed by one-way ANOVA with Fisher’s least significant difference (*LSD*) post-hoc test using SPSS statistics 20 software (version 20). Significance was assessed at *p* < 0.05.

## 3. Results

### 3.1. Fluoride Increased the Amount of p21 mRNA and the Amount of Phosphorylated p21

Fluoride treatment increased acetylated-p53 at Lys379 (Ac-p53) in LS8 cells (Appendix A) and in rat ameloblasts [19]. Ac-p53 can induce p21 transcription and once p21 is phosphorylated (p-p21) at Thr145, cell survival is enhanced. NaF (5 mM) significantly increased *p21* mRNA levels at 24 h compared to control (0 mM) (** *p* < 0.01) (Figure 1a). At early time points (1–6 h), NaF increased p-p21 protein expression (Figure 1b,c), which can counteract fluoride-induced apoptosis. However, at 24 h, p21 and p-p21 protein levels were decreased with fluoride treatment (Figure 1c), even though *p21* mRNA levels were significantly increased at 24 h (Figure 1a). These results suggest that fluoride may facilitate p21 protein degradation after 24 h of treatment.

### 3.2. Proteasomal Inhibitor MG-132 Reversed Fluoride-Mediated p21 Protein Attenuation and Alleviated Apoptosis

Next, we investigated the role of the proteasome in fluoride-mediated p21 protein levels. NaF (5 mM) treatment for 6 h increased Ubiquitinated-p21 (Ub-p21) in LS8 cells (Figure 2a), indicating that fluoride induces p21 ubiquitin-proteasomal degradation. Figure 2b,c show that proteasome inhibitor MG-132 (0.5 and 1 μM) reversed fluoride-mediated p21 protein degradation (Figure 2b) and increased p-p21 levels at 24 h (Figure 2b). Figure 2c shows that cytoplasmic p-p21 was increased by MG132 (0.5 μM) compared to fluoride alone at 24 h.

MG-132 (1 μM) treatment alone significantly increased *p21* mRNA compared to control (Appendix A), whereas MG-132 did not increase fluoride-induced *p21* mRNA levels (Appendix A). The results presented in Appendix A suggest that MG-132 reverses fluoride-mediated p21 protein degradation independently of *p21* transcription. Previously we demonstrated that fluoride induced caspase-3 cleavage and DNA fragmentation and in LS8 cells [12,16]. So, we assessed MG-132 effect on fluoride-induced apoptosis in LS8 cells. NaF (5 mM) significantly decreased the *Bcl-2*/*Bax* mRNA ratio compared to control (*p* < 0.01) (Figure 3a). This was significantly reversed by addition of 1 μM MG-132 (*p* < 0.05) (Figure 3b). In addition, fluoride-induced caspase-3 cleavage and fluoride-induced expression of the DNA damage marker γH2AX were suppressed by MG-132 treatment (Figure 3c). Therefore, MG-132 attenuated fluoride-induced apoptosis and this correlated with upregulation of p21 and p-p21 protein in LS8 cells. These results suggest that p21 proteasomal degradation plays a critical role in promoting fluoride-induced apoptosis.

### 3.3. Fluoride-Induced Expression of Mdm2 mRNA and p-MDM2 Protein

MDM2 (ubiquitin ligase E3) binds to p53 and inhibits the p53 pathway by promoting ubiquitination of p53 (Ub-p53) to initiate p53 proteasomal degradation [26]. Likewise, MDM2 binds to p21 and increases p21 proteasomal degradation [27]. After LS8 cells were treated with NaF at the indicated concentrations for 24 h, *Mdm2* mRNA levels were significantly increased by treatment with 3 mM or 5 mM NaF (*p* < 0.01) (Figure 4a). p-MDM2 [Ser166] increases MDM2-mediated degradation of p53 [28] or p21 [29]. So, we asked if fluoride increases p-MDM2 in vitro and in vivo. Western blot results show that in LS8 cells p-MDM2 protein levels were increased by NaF (5 mM) treatment after 1 h to 24 h (Figure 4b). Figure 5 shows p-MDM2 Immunohistochemistry (IHC) staining of mouse incisors treated with fluoride (0 or 150 ppm) for 6 weeks. Fluoride treatment dramatically increased p-MDM2 levels in mouse ameloblasts when compared to control ameloblasts. Note that secretory stage (SEC) ameloblasts treated with 150 ppm fluoride were torn due to a sectioning artifact (Figure 5b). These results suggest that following acetylation of p53, fluoride increased expression of *Mdm2* mRNA and p-MDM2 protein that promote MDM2-mediated proteasomal degradation of p53 and p21.

### 3.4. Fluoride-Induced MDM2-p53 Binding and Increased Ubiquitination of p53

Next, we investigated whether fluoride affects MDM2-p53 protein binding and p53 ubiquitination. The co-IP results show that NaF (5 mM) treatment for 6 h increased the MDM2-p53 protein interaction and induced Ub-p53 levels in LS8 cells. (Figure 6a). MDM2 antagonist, Nutlin-3a (5 μM) inhibited the fluoride-induced MDM2-p53 interaction and significantly decreased Ub-p53 levels compared to fluoride alone (Figure 6b). These results provide further evidence that MDM2-mediated p53 ubiquitination directs p53 proteasomal degradation.

### 3.5. Inhibition of MDM2-p21 Formation by Nutlin-3a Ameliorated the Fluoride-Mediated p21/p-p21 Decrease

MDM2 binds to p21 and promotes p21 proteasomal degradation [27]. We assessed MDM2 involvement in the fluoride-mediated decrease of p21 protein levels observed at 24 h. The co-IP result shows that NaF (5 mM) treatment for 6 h increased MDM2-p21 binding which was suppressed by Nutlin-3a (Figure 7a). Nutlin-3a reversed the fluoride-mediated p21 protein degradation at 24 h (Figure 7b) and increased p-p21 (Figure 7b,c). These results suggest that fluoride decreases p21 protein levels through MDM2-p21 binding. Nutlin-3a (5 μM) alone significantly increased *p21* mRNA levels in LS8 cells when compared to untreated controls (0 μM) after 24 h treatment (Appendix A). However, Nutlin-3a did not alter fluoride-induced *p21* mRNA levels (Figure 7d), indicating that Nutlin-3a reversed p21 protein attenuation independently of *p21* transcription. These results suggest that fluoride promoted MDM2-mediated p21 proteasomal degradation.

### 3.6. Nutlin-3a Attenuated Fluoride-Induced Apoptosis

Since Nutlin-3a increased p21 and p-p21 (Figure 7b,c), and since p21 and p-p21 can counteract apoptosis we investigated how Nutlin-3a affects fluoride-induced apoptosis. NaF (5 mM) treatment for 24 h induced the DNA damage marker γH2AX and also induced caspase-3 cleavage in LS8 cells. Nutlin-3a reversed the fluoride- mediated γH2AX induction and inhibited caspase-3 cleavage (Figure 8). These data suggest that MDM2-mediated p21 proteasomal degradation plays a pro-apoptotic role in fluoride toxicity.

## 4. Discussion

Previously we reported that fluoride increased Ac-p53 levels. Ac-p53 participated in LS8 cell apoptosis in vitro and also participated in ameloblast apoptosis in vivo [19]. Fluoride-induced apoptosis is not specific to ameloblasts. However, ameloblasts are very sensitive to stress. A high fever can affect ameloblasts and result in malformed enamel. The process of being born can cause a defect in the enamel termed the “neonatal line” and ameloblasts are also more sensitive to fluoride toxicity than are other cells [33]. Here, we investigated Ac-p53 downstream pathways that contribute to fluoride toxicity. Figure 9 shows a schema of MDM2-p53 and MDM2-p21 signaling during fluoride toxicity. Our results showed that fluoride induced p21 and p-p21 protein levels and that these proteins play a protective role against fluoride-induced apoptosis in the early phase (6 h). Conversely, MDM2-mediated p21 proteasomal degradation in the late phase (24 h) resulted in fluoride toxicity (Figure 9). p21 is one of the transcriptional targets of p53 [19]. p21 can function as cell cycle arrest, transcriptional regulation and anti-apoptotic factor depending on cell type and cellular conditions [34]. p21 function can be modulated not only at the transcriptional level but also at the post-translational level by phosphorylation. Akt-mediated phosphorylation of p21 at Thr145 (p-p21) induces p21 translocation from the nucleus to the cytosol [23]. In the cytosol, p21 can inhibit the pro-apoptotic kinase ASK1 through direct interaction [35], and p21 binds to procaspase-3, blocking its proteolytic activation to promote cell survival [24]. In the present study, fluoride increased p-p21 at the early phase (from 1 to 6 h) (Figure 1b,c). This transient increase of p-p21 can counteract fluoride-induced apoptosis. However, fluoride-induced p-p21 and p21 protein levels were reduced at 24 h (Figure 1c) even though *p21* mRNA levels were significantly increased by 24 h of fluoride treatment (Figure 1a). To elucidate the discrepancy between *p21* mRNA levels and p21 protein levels at 24 h, we asked if the proteasome and MDM2 play a role in the fluoride-mediated p21 protein reduction.

The 26S proteasome, a multicatalytic enzyme complex, is the main intracellular proteolytic system involved in the degradation of ubiquitinated (Ub) proteins [36]. MG-132 is a potent cell-permeable proteasome inhibitor and non-specifically leads to stabilization of multiple different proteins, including p53. Fluoride treatment for 24 h increased Ub-p21 that then decreased p21 protein levels. MG-132 reversed p21 attenuation at 24 h and increased p-p21 levels (Figure 2), indicating that fluoride promotes p21 ubiquitin-proteasomal degradation. While MG132+fluoride treatment increased total protein ubiquitination compared to fluoride alone, we observed that fluoride-induced Ub-p21 was decreased by MG132 treatment. Our result is consistent with a recent report showing that MG132 decreased Ub-p21 by inducing the Ubiquitin-specific processing protease (USP)11, which is a deubiquitylase that directly removes p21 polyubiquitylation and stablilizes the p21 protein [37]. Although MG-132 treatment (1 μM) alone significantly increased *p21* mRNA expression (Appendix A), MG-132 did not significantly alter *p21* mRNA levels after fluoride treatment (Appendix A). These results suggest that MG-132 reversed the fluoride-mediated p21 protein reduction via inhibition of p21 proteasomal degradation and this occurred independently of *p21* transcription. MG-132 can be effective in cancer treatment by inducing apoptosis in tumor cells [38,39]. In contrast, accumulating data demonstrate that MG-132 can protect cells and tissues from oxidative damage [40]. During oxidative damage, MG-132 can activate the Nrf2 signaling pathway that upregulates anti-apoptotic factor Bcl-2 to prevent apoptosis [41]. In the present study, MG-132 significantly increased the *Bcl2/Bax* mRNA ratio in fluoride treated cells (Figure 3b) and MG-132 increased p21/p-p21 protein levels (Figure 2b,c) to mitigate fluoride-induced apoptosis (Figure 3c). These results coincide with a previous study showing that MG-132 blocks ultraviolet (UV)-induced apoptosis and that this correlates with p53 stabilization and upregulation of p21 [42].

MDM2 (ubiquitin ligase E3) is one of the transcriptional targets of p53 [25]. Fluoride increased *Mdm2* mRNA levels, and increased p-MDM2 [Ser166] protein levels in vitro (Figure 4) and also p-MDM2 [Ser186] in vivo (Figure 5). p-MDM2 [Ser166 or Ser186] promotes p53 and p21 proteasomal degradation [28,29]. Our results showed that MDM2 directly bound to p21 after fluoride treatment (Figure 7), suggesting that fluoride promotes MDM2-mediated p21 proteasomal degradation and therefore promotes apoptosis.

Nutlin-3a is an MDM2 antagonist that binds in the p53-binding pocket of MDM2 to inhibit MDM2-mediated p53 proteasomal degradation [43]. Intriguingly, our results showed that Nutlin-3a inhibited both MDM2-p53 binding (Figure 6b) and MDM2-p21 binding (Figure 7a). Since Nutlin-3a occupies the N-terminal p53-binding pocket of MDM2, Nutlin-3a can also interfere with other proteins that bind MDM2 at or near this same pocket [44]. In the present study, Nutlin-3a inhibited MDM2-p21 binding (Figure 7a) to reverse p21 protein degradation and increase p-p21 protein levels (Figure 7b), indicating that MDM2 also promotes p21 proteasomal degradation during fluoride toxicity. Although Nutlin-3a alone significantly increased *p21* mRNA expression (Appendix A), Nutlin-3a did not alter fluoride-induced *p21* mRNA levels (Figure 7c). These results suggest that Nutlin-3a reversed the fluoride-mediated p21 protein attenuation through a *p21* transcription independent manner. Nutlin-3a can induce or inhibit apoptosis [45,46], depending on cell type, tissue and circumstances. In melanoma cells, loss of p21 promoted drug-induced DNA damage and Nutlin-3a protected cells from DNA damage via p53-dependent activation of p21 [21]. This report is in concordance with our results which showed that Nutlin-3a increased p21 and p-p21 levels to suppress fluoride-induced caspase-3 cleavage and expression of the DNA damage marker γH2AX (Figure 8). These results suggest that the MDM2-mediated suppression of p-p21 and p21 plays a role in fluoride cytotoxicity. In contrast to MG-132 (Figure 3b), the MDM2 antagonist Nutlin-3a did not alter the *Bcl2/Bax* mRNA ratio during fluoride treatment (Appendix A), indicating that Nutlin-3a acts primarily by inhibiting MDM2-mediated proteasomal degradation during fluoride toxicity. In contrast, the proteasome inhibitor MG-132 did increase the *Bcl2/Bax* ration that was reduced after fluoride treatment, indicating that the proteasome plays a role in fluoride-mediated apoptosis. However, both MG-132 and Nutlin-3a inhibited caspase-3 cleavage, indicating some overlap of function between the two inhibitors.

## 5. Conclusions

This is the first report implicating MDM2-p53 and MDM2-p21 signaling pathways in fluoride toxicity. MDM2-mediated p21 proteasomal degradation plays a critical role as a pro-apoptotic factor during fluoride toxicity in ameloblast-derived LS8 cells. The MDM2-p53 and MDM2-p21 pathways may be potential therapeutic targets for fluoride-mediated health problems.

## Figures and Tables

**Figure 1 cells-08-00436-f001:**
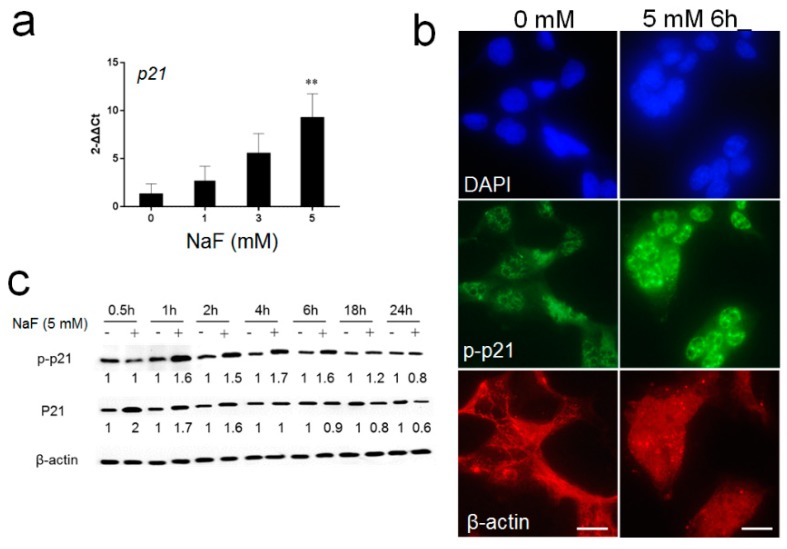
Fluoride upregulates *p21* mRNA and p21/p-p21 protein in LS8 cells. (**a**) LS 8 cells were treated with NaF at the indicated concentrations for 24 h and *p21* mRNA was then quantified by quantitative polymerase chain reaction (qPCR). Fluoride (5 mM) treatment significantly increased *p21* mRNA levels. *Gapdh* was the internal reference control gene. Data are presented as means ± standard deviation (SD) (** *p* < 0.01 vs. 0 mM). (**b**) Cells were treated with NaF (5 mM) for 6 h and phospo-p21 (p-p21; green), nucleus (4′,6-diamidino-2-phenylindole (DAPI); blue) and β-actin (red) were detected by immunocytochemistry. Fluoride increased p-p21 protein levels in LS8 cells. (**c**) LS8 cells were treated with NaF (5 mM) for the indicated times and p21 (18 kDa) and p-p21 (21 kDa) were detected by Western blots. Fluoride increased p21 and p-p21 protein expression in the early phase (1 to 6 h), which then decreased at 24 h. The numbers show the relative expression normalized by the loading control β-actin (44 kDa). Statistical analysis of relative protein expression of p21 and p-p21 are shown in Appendix A.

**Figure 2 cells-08-00436-f002:**
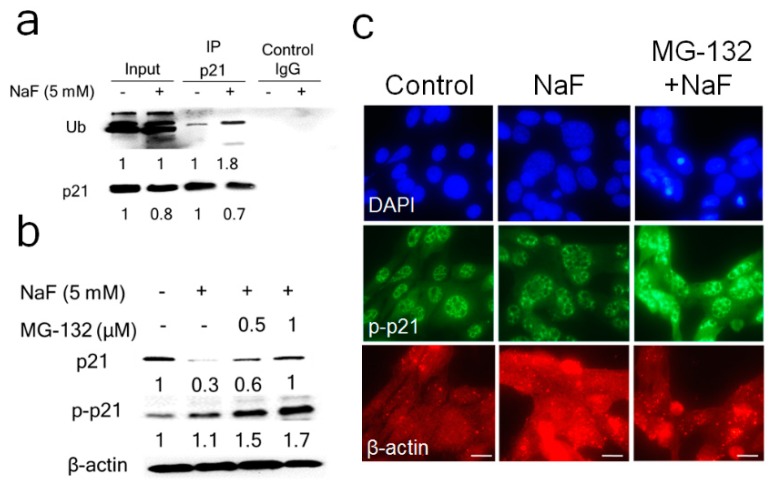
Fluoride increased Ub-p21 binding and MG-132 reversed the fluoride-mediated p21 protein decrease. (**a**) LS8 cells were treated with NaF (5 mM) for 6 h. Protein was immunoprecipitated using anti-p21 antibody and ubiquitinated-p21 (Ub-p21) was detected in the precipitated fraction by the anti-Ubiquitin antibody. Fluoride treatment increased Ub-p21 levels. IgG was used as the negative control. The numbers show relative protein expression vs. Controls (0 mM NaF). IP lanes were quantified separately from input lanes. (**b**) LS8 cells were treated with MG-132 (0.5–1.0 μM) for 2 h prior to NaF (5 mM) treatment for 24 h. p21 (18 kDa) and p-p21 (21 kDa) were detected by Western blot. MG-132 reversed the fluoride-induced p21 suppression at 24 h by increasing p-p21 protein levels. The numbers show relative expression normalized by the loading control β-actin (44 kDa). Statistical analysis of relative protein expression of p21 and p-p21 are shown in Appendix A. (**c**) Cells were treated with NaF (5 mM) with/without MG132 (0.5 µM) for 24 h and p-p21 (green), nucleus (DAPI; blue) and β-actin (red) were detected by immunocytochemistry. MG-132 treatment increased p-p21 protein levels.

**Figure 3 cells-08-00436-f003:**
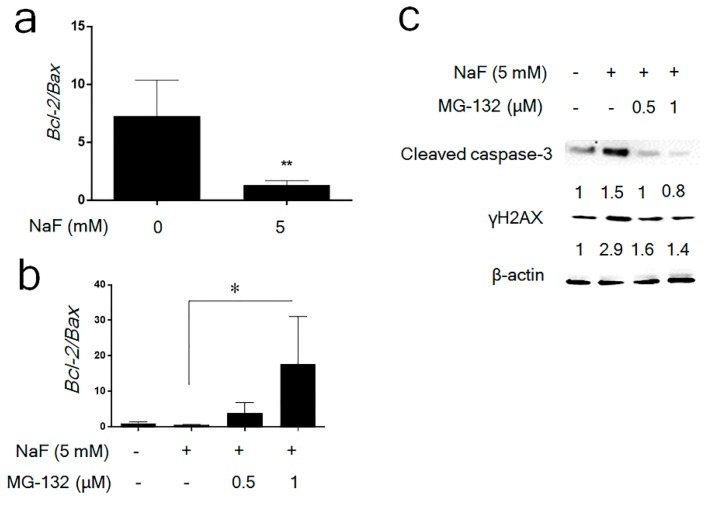
MG-132 attenuated fluoride-induced apoptosis in LS8 cells. LS8 cells were treated with MG-132 (0.5–1.0 μM) for 2 h prior to NaF (5 mM) treatment for 24 h. (**a**) The *Bcl-2/Bax* mRNA ratio was quantified by qPCR. NaF significantly decreased the *Bcl-2/Bax* mRNA ratio (** *p* < 0.01). (**b**) MG-132 (1 μM) significantly increased the *Bcl-2/Bax* ratio compared to NaF treatment alone (* *p* < 0.05). Data are presented as means ± SD. (**c**) γH2AX (15 kDa) and cleaved-caspase-3 (17 kDa) were detected by Western blots. MG-132 inhibited fluoride-induced γH2AX protein expression and inhibited caspase-3 cleavage. The numbers show relative expression normalized by the loading control β-actin (44 kDa). Statistical analysis of relative protein expression of cleaved-caspase-3 and γH2AX are shown in Appendix A.

**Figure 4 cells-08-00436-f004:**
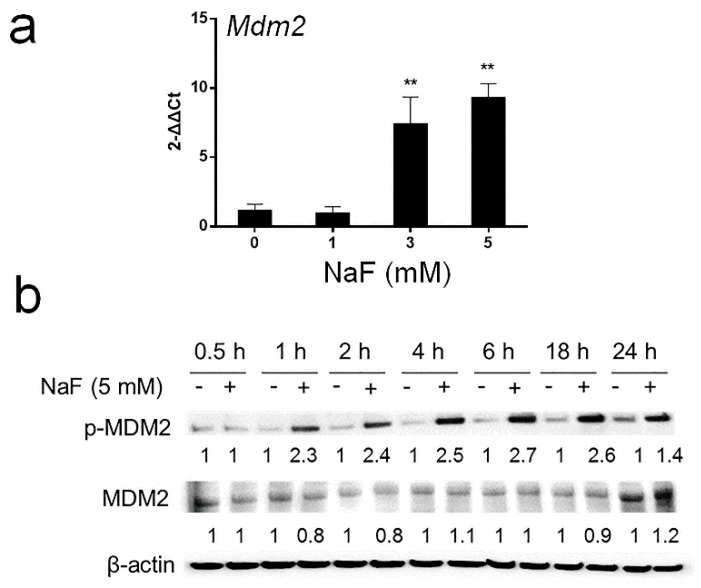
Fluoride induced *Mdm2* mRNA expression and induced p-MDM2 protein levels in LS8 cells. (**a**) LS8 cells were treated with the indicated concentrations of NaF for 24 h and *Mdm2* mRNA was quantified by qPCR. Fluoride (3 mM or 5 mM) significantly increased *Mdm2* expression. *Gapdh* was the internal reference control gene. Data are presented as the mean ± SD (** *p* < 0.01 vs. 0 mM). (**b**) Cells were treated with NaF (5 mM) for the indicated times. Whole cell lysates were subjected to Western blot analysis for phospho-MDM2 (p-MDM2 [Ser166]) (90 kDa) and total MDM2 (MDM2) (90 kDa) expression. β-actin (44 kDa) was used as a loading control. The numbers show relative protein expression normalized by the β-actin loading control. Statistical analysis of relative protein expression of MDM2 and p-MDM2 are shown in Appendix A.

**Figure 5 cells-08-00436-f005:**
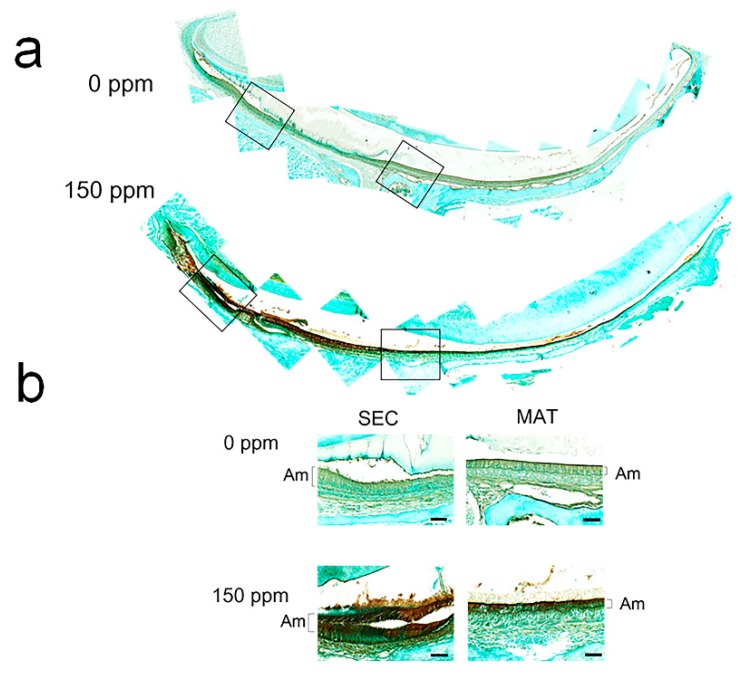
Fluoride increased p-MDM2 protein expression in mouse ameloblasts. Mice were treated with 0 or 150 ppm fluoride in drinking water for 6 weeks. (**a**) p-MDM2 [Ser185] was detected by immunohistochemistry in maxillary incisors from mice treated with 0 ppm (upper panel) or 150 ppm (lower panel) fluoride. (**b**) High magnification of p-MDM2 staining in secretory stage (SEC) and maturation stage (MAT) mouse enamel organs. More p-MDM2 was formed in mouse ameloblasts treated with 150 ppm fluoride compared to control ameloblasts (0 ppm). Shown are representative images from three mice. Scale bar represents 20 μm. Brackets denote ameloblasts (Am).

**Figure 6 cells-08-00436-f006:**
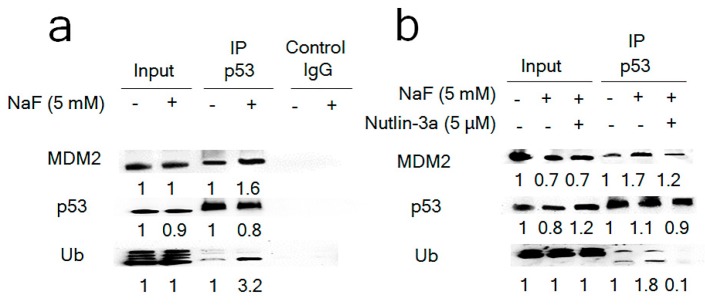
Fluoride induced p53-MDM2 binding and p53 ubiquitination in LS8 cells. LS8 cells were treated with fluoride for 6 h and protein was immunoprecipitated using anti-p53 antibody. p53, MDM2 and ubiquitin were detected by Western blot. (**a**) NaF increased MDM2-p53 binding and increased amounts of ubiquitinated-p53 (Ub-p53). (**b**) MDM2 antagonist Nutlin-3a suppressed fluoride-induced MDM2-p53 binding and decreased Ub-p53 levels. Control IgG was used as a negative control. The numbers show relative protein expression vs. Control (0 mM NaF). Input and Immunoprecipitation (IP) lane relative expression was calculated separately.

**Figure 7 cells-08-00436-f007:**
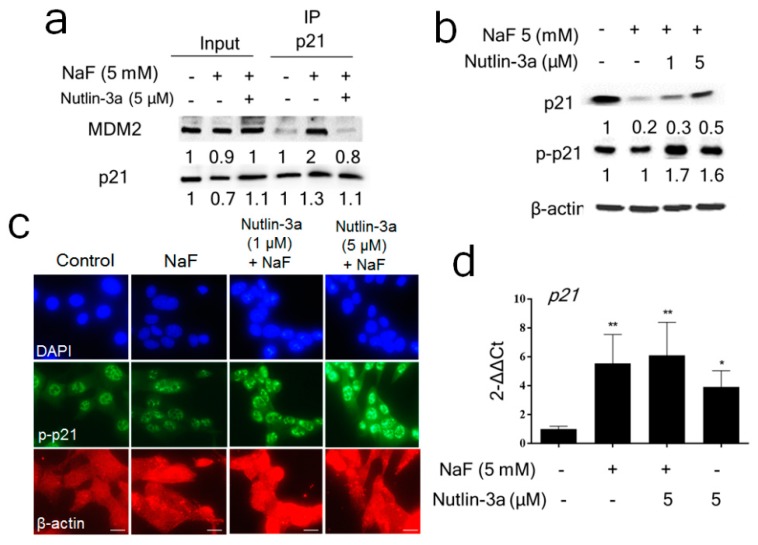
Nutlin-3a reversed the fluoride-mediated p21 protein decrease by increasing p-p21 levels. (**a**) LS8 cells were treated with Nutlin-3a (5 μM) for 2 h followed by the additional NaF (5 mM) for 6 h and then protein was immunoprecipitated using anti-p21 antibody. NaF treatment increased MDM2-p21 binding and Nutlin-3a inhibited this binding. The numbers show relative protein expression vs. control (0 mM NaF). Input and IP lane relative expression was calculated separately for IP lanes. (**b**) LS8 cells were treated with Nutlin-3a (1–5 μM) for 2 h followed by the additional NaF (5 mM) for 24 h. p21 (18 kDa) and p-p21 (21 kDa) were detected by Western blots. Nutlin-3a treatment (24 h) reversed fluoride-mediated p21 degradation and increased p-p21 levels. The numbers show relative expression normalized by the loading control β-actin (44 kDa). Statistical analysis of relative protein expression of p21 and p-p21 are shown in Appendix A. (**c**) Cells were treated with fluoride (5 mM) with/without Nutlin-3a for 24 h. p-p21 (green), nucleus (DAPI; blue) and β-actin (red) expression were detected by immunocytochemistry. Nutlin-3a addition augmented p-p21 expression compared to NaF treatment alone. (**d**) LS8 cells were treated with Nutlin-3a (5 μM) for 2 h followed by the additional NaF (5 mM) for 24 h. Nutlin-3a alone significantly increased *p21* mRNA compared to controls, but Nutlin-3a treatment with NaF did not alter *p21* expression compared to NaF alone. Data are presented as the mean ± SD (**; *p* < 0.01, *; *p* < 0.05 vs. 0 mM).

**Figure 8 cells-08-00436-f008:**
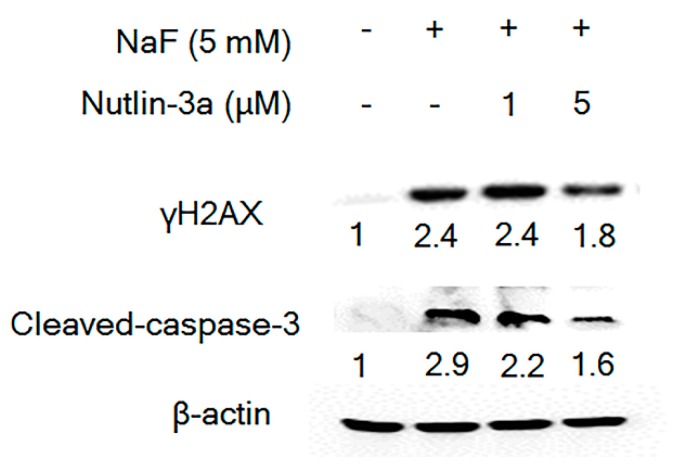
Nutlin-3a ameliorated fluoride-induced apoptosis in LS8 cells. LS8 cells were treated with Nutlin-3a (1 μM or 5 μM) for 2 h followed by the addition of NaF (5 mM) for 24 h. DNA damage marker γH2AX (15 kDa) expression and caspase-3 cleavage (17 kDa) were detected by Western blot. Nutlin-3a attenuated caspase-3 cleavage and reduced γH2AX expression. The numbers show relative expression normalized by the loading control β-actin (44 kDa). Statistical analysis of relative protein expression of cleaved-caspase-3 and γH2AX are shown in Appendix A.

**Figure 9 cells-08-00436-f009:**
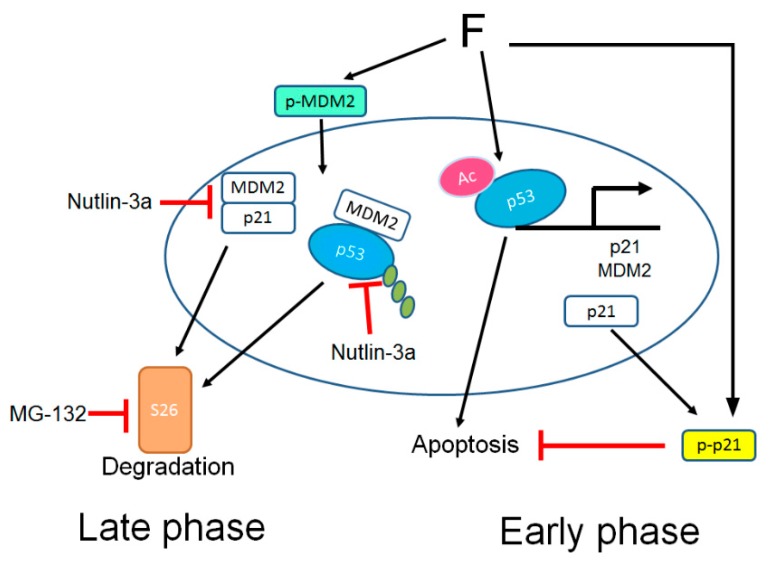
Schema of MDM2-p53 and MDM2-p21 signaling in fluoride toxicity. Fluoride increases acetylated-p53 (Ac-p53) levels to upregulate transcription of *Mdm2* and *p21*. In the early phase (1–6 h), fluoride induces phosphorylation of p21 (p-p21), which translocates p21 from the nucleus to the cytoplasm where p-p21 counteracts fluoride-induced apoptosis. Fluoride enhances MDM2-p53 and MDM2-p21 formation to promote MDM2-mediated p53 and p21 proteasomal degradation that leads to p21 and p-p21 attenuation in the late phase (24 h). Nutlin-3a inhibits MDM2-p53 and MDM2-p21 binding. Nutlin-3a or MG-132 (proteasome inhibitor) reverses fluoride-induced p21 attenuation and increases p-p21 to suppress fluoride-mediated apoptosis.

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
