# Peer review of "MDM2-Mediated p21 Proteasomal Degradation Promotes Fluoride Toxicity in Ameloblasts"

_cells, 2019, doi:10.3390/cells8050436_

Round 1

Reviewer 1 Report

The manuscript is well written and clearly presented.

Minor comment: The name of the software should be mentioned under statistical analysis section.

Author Response

Minor comment: The name of the software should be mentioned under statistical analysis section.

Revision: We thank the reviewer for stating that: The manuscript is well written and clearly presented.

We added the software name (SPSS statistics 20) in M&M section.

Reviewer 2 Report

In this manuscript, authors investigated the contributions of p21 and MDM2 to fluoride toxicity in LS8 ameloblast cell line.  They found that chronic exposure to 5 mM NaF induced apoptosis and p21 reduction, which were reversed in the presence of MG-132 (proteasome inhibitor) or Nutlin-3a (MDM2 antagonist). Although those results are interesting, there is a concern in the reproducibility as most of findings likely relay on single experiment through Western blotting. At least three independent experiments should be conducted with proper statistical analyses. Additional comments are as follow;

1. Since high concentration of sodium fluoride was used in the study, some of results may be experimental artifacts. Do potassium fluoride and sodium chloride have different effects on ameloblasts?

2. It is unclear if the phenotypes are specific to ameloblasts. Other cell types should be also used to validate their findings.

3. While administrations with a MDM2 antagonist or proteasome inhibitor reversed NaF-induced phenotypes, there is no direct evidence showing that p21 proteasomal degradation is the predominant pathway.

Author Response

Comment: Although those results are interesting, there is a concern in the reproducibility as most of findings likely relay on single experiment through Western blotting. At least three independent experiments should be conducted with proper statistical analyses.

Revision: We did independent experiments at least three times for western blotting and confirmed the reproducibility. Representative images are shown in each figure. We added results of statistical analysis for each western blot experiment in supplementary materials (Fig.S4 to Fig.S9).

Additional comment 1. Since high concentration of sodium fluoride was used in the study, some of

results may be experimental artifacts. Do potassium fluoride and sodium chloride have different effects on ameloblasts?

Revision: KF can cause dental and skeletal fluorosis. Animal models demonstrated that potassium fluoride affected ameloblasts resulting in enamel malformation (Cheyne VD, https://doi.org/10.1177/00220345420210020501). (Greenwood DA, https://digitalcommons.usu.edu/honor_lectures/41). We observed sodium chloride (NaCl) at the same concentrations of NaF (5-10 mM) did not affect ameloblast-derived L8 cells in vitro (Suzuki et al, Doi:10.1016/j.freeradbiomed.2015.08.015). We have added this explanation to the Discussion section.

Additional comment 2. It is unclear if the phenotypes are specific to ameloblasts. Other cell types

should be also used to validate their findings.

Revision: The phenotype is not specific to ameloblasts. However, ameloblasts are very sensitive to stress. A high fever can affect ameloblasts and result in malformed enamel. The process of being born can cause a defect in the enamel termed the “neonatal line” and ameloblasts are also more sensitive to fluoride toxicity than are other cells (Sharma R et al. The acid test of fluoride: how pH modulates toxicity. PLoS One. 2010 May 28;5(5):e10895. doi: 10.1371/journal.pone.0010895 PMID: 20531944). We have now described this in the Discussion section.

Additional comment 3. While administrations with a MDM2 antagonist or proteasome inhibitor

reversed NaF-induced phenotypes, there is no direct evidence showing that p21 proteasomal degradation is the predominant pathway.

Revision: Our results show that fluoride increased ubiquitination of p21 (Fig.2a) that suggests at least one pathway that eliminates p21 and it is the predominant pathway for all protein elimination.

Reviewer 3 Report

In this work, the authors have investigated the mechanism of fluoride toxicity. Based on their previous work that indicated increased p53 signaling upon fluoride treatment, the authors wanted to further investigate the role of the downstream signaling molecule, p21 in this toxicity. Based on their experiments, the authors demonstrate that while p21 transcripts are upregulated upon NaF treatment, p21 protein is increased at shorter timescales (~6hr) wherein p21 is phosphorylated and localizes to the cytoplasm (!) where it supposedly inhibits caspase activation and therefore inhibits apoptosis. However, p21 protein levels are reduced at the 24hr mark, thereby leading to increased caspase activation and apoptosis. The reduction in p21 at 24hr point is dependent on MDM2-mediated ubiquitination followed by proteasomal degradation of p21 protein. To support the role of MDM2 and the proteasome, the authors utilize MDM2 inhibitor and a proteasome inhibitor and show a stabilization of p21 upon treatment with either of these two proteins.

Overall, the logic of the experiments is sound and conclusions are mostly supported by their data. Although, I have a few recommendations below that will bolster their findings.

Fig 1B: increased p21 is mostly nuclear. Images are not clear enough to show the distinct increase in cytoplasmic p21. How about showing a more zoomed in higher quality image for p21 at 0, 6 and 24h.

Fig 2a: Authors claim that p21 is ubiquitinated based on the Co-IP experiment indicating pull down of ubiquitin along with p21. This is a weak experiment and the authors should at the very least do a co-IP with ubiquitin to demonstrate a pull down of p21 and/or a p21 western showing increased number of slower-migrating bands upon MG132+NaF treatment.

Fig 2C: Images are out of focus, especially the DAPI channel. The images are also overexposed for both DAPI and the p21 channel showing non-specific cytoplasmic signals. This figure is not cited in the results section.

Fig 3: MG132 is a proteasomal inhibitor and will non-specifically lead to stabilization of multiple different proteins, including p53.

Fig 7: The IF images add no value since the reader can barely see the cellular distribution of the p21 protein. Please provide, larger, higher quality and more zoomed in images of p21. Since in these miniature images too, one can easily see that p21 distribution is not similar to the other cytoplasmic protein, B-actin, there is no support to the authors’ claim of p21 being cytoplasmic. I’d recommend repeating the IF (along with a siRNA control to show the antibody specificity) and performing cellular fractionation followed by western blot analysis.

Other point: If NaF treatment leads to increased p-MDM2 levels and therefore reduced p53 protein (via MDM2 mediated ubiquitination), why is there no decrease in p21 transcription (supplementary figure 2) with the same treatment?

Author Response

Recommendations 1.  Fig 1B: increased p21 is mostly nuclear. Images are not clear enough to show the distinct increase in cytoplasmic p21. How about showing a more zoomed in higher quality image for p21 at 0, 6 and 24h.

Revision: Figure1B images were replaced with more zoomed pictures to show the increase in cytoplasmic p-p21.

Recommendations 2.  Fig 2a: Authors claim that p21 is ubiquitinated based on the Co-IP experiment

indicating pull down of ubiquitin along with p21. This is a weak experiment and the authors should at the very least do a co-IP with ubiquitin to demonstrate a pull down of p21 and/or a p21 western showing increased number of slower-migrating bands upon MG132+NaF treatment.

Revision:

While MG132+fluoride treatment increased total protein ubiquitination compared to fluoride alone, we observed that fluoride-induced Ub-p21 was decreased by MG132 treatment. Our result is consistent with a recent report showing that MG132 decreased Ub-p21 by inducing the Ubiquitin-specific processing protease (USP)11, which  is a deubiquitylase that directly removes p21 polyubiquitylation and stablilizes the p21 protein (Deng T 2019 DOI: 10.1073/pnas.1714938115) . This explanation is now present in our Discussion section.

Recommendations 3. Fig 2C: Images are out of focus, especially the DAPI channel. The images are

also overexposed for both DAPI and the p21 channel showing non-specific cytoplasmic signals. This figure is not cited in the results section.

Revision: Fig 2C images were replaced with more zoomed in pictures. We have now stated in the result section that MG132 increased p-p21 in the cytoplasm compared to NaF alone.

Recommendations 4. Fig 3: MG132 is a proteasomal inhibitor and will non-specifically lead to

stabilization of multiple different proteins, including p53.

Revision: We have now stated in the discussion section that MG132 is a proteasomal inhibitor and will non-specifically lead to stabilization of multiple different proteins, including p53.

Recommendations 5. Fig 7: The IF images add no value since the reader can barely see the cellular

distribution of the p21 protein. Please provide, larger, higher quality and more zoomed in images of p21. Since in these miniature images too, one can easily see that p21 distribution is not similar to the other cytoplasmic protein, B-actin, there is no support to the authors’ claim of p21 being cytoplasmic. I’d recommend repeating the IF (along with a siRNA control to show the antibody specificity) and performing cellular fractionation followed by western blot analysis.

Revision: Fig 7C images were replaced with more zoomed pictures to show Nutlin-3a increased p-p21 in the cytoplasm compared to NaF alone.

Other point: If NaF treatment leads to increased p-MDM2 levels and therefore reduced p53 protein (via MDM2 mediated ubiquitination), why is there no decrease in p21 transcription (supplementary figure 2) with the same treatment?

Revision: p21 transcription can be regulated by p53-independent mechanisms (Karimian A, http://dx.doi.org/10.1016/j.dnarep.2016.04.008). Therefore p21 transcription levels may not correlate with p53 levels especially when considering all the other functions of p21.

Round 2

Reviewer 2 Report

Authors have sufficiently revised the manuscript.